# Online Algorithms for Multi-shop Ski Rental with Machine Learned Advice

**Shufan Wang**
Binghamton University
State University of New York
Binghamton, NY 13902
swang214@binghamton.edu

**Jian Li**
Binghamton University
State University of New York
Binghamton, NY 13902
lij@binghamton.edu

**Shiqiang Wang**
IBM Thomas J. Watson Research Center
Yorktown Heights, NY 10598
wangshiq@us.ibm.com

## Abstract

We study the problem of augmenting online algorithms with machine learned (ML) advice. In particular, we consider the *multi-shop ski rental* (MSSR) problem, which is a generalization of the classical ski rental problem. In MSSR, each shop has different prices for buying and renting a pair of skis, and a skier has to make decisions on when and where to buy. We obtain both deterministic and randomized online algorithms with provably improved performance when either a single or multiple ML predictions are used to make decisions. These online algorithms have no knowledge about the quality or the prediction error type of the ML prediction. The performance of these online algorithms are robust to the poor performance of the predictors, but improve with better predictions. Extensive experiments using both synthetic and real world data traces verify our theoretical observations and show better performance against algorithms that purely rely on online decision making.

## 1 Introduction

Uncertainty plays a critical role in many real world applications where the decision maker is faced with multiple alternatives with different costs. These decisions arise in our daily lives, such as whether to rent an apartment or buy a house, which cannot be answered reliably without knowledge of the future. In a more general setting with multiple alternatives, such as a large number of files with different execution times in a distributed computing system, it is hard to decide which file should be executed next without knowing which file will arrive in the future. These decision-making problems are usually modeled as online rent-or-buy problems, such as the classical *ski rental* problem and many of its generalizations [5, 6, 7, 8, 11, 15].

Two paradigms have been widely studied to deal with such uncertainty. On the one hand, online algorithms are designed without prior knowledge to the problem, and *competitive ratio* (CR) is used to characterize the goodness of the algorithm in lack of the future. CR is defined as the ratio between the cost of the online algorithm (ALG) and that of the offline optimal (OPT), in the worst-case over all feasible inputs. On the other hand, machine learning is applied to address uncertainty by making future predictions via building robust models on prior data. Recently, there is a popular trend in the design of online algorithms by incorporating machine learned (ML) predictions to improving their performance [2, 3, 4, 9, 10, 12, 14, 16, 17, 18]. To that end, two properties are desired: (i) if the

predictor is good, the online algorithm should perform close to the best offline algorithm (a design goal called *consistency*); and (ii) if the predictor is bad, the online algorithm should not degrade significantly, i.e., its performance should be close to the online algorithm without predictions (a design goal called *robustness*). Importantly, these properties are achieved under the assumption that the online algorithm has no knowledge about the quality of the predictor or the prediction error types.

**The multi-shop ski rental problem.** While previous studies focused on using ML predictions for a skier to buy or rent the skis in a single shop, we study the more general setting where the skier has multiple shops to buy or rent the skis with different buying and renting prices. We call this a *multi-shop ski rental* (MSSR) problem. This is often the case in practice where the skier has to make a *two-fold* decision, i.e, *when and where to buy*. The MSSR not only naturally extends the classical ski rental problem, where the skier rents or buys the skis in a single shop, but also allows heterogeneity in skier's options. This desirable feature makes the ski rental problem a more general modeling framework for online algorithm design. A few real world applications that can be modeled with MSSR are presented in the supplementary material.

Furthermore, we consider not only the case of using prediction inputs from a *single* ML algorithm, but also a more general setting where predictions are drawn from multiple ML models. Closest to ours is [4] where multiple experts provide advice in a single shop, which can be considered as a special case of ours. However, we incorporate multiple predictions into decision making by comparing the number of predictions to a threshold, which is much easier to implement in real applications.

**Consistency and Robustness.** Inspired by [12, 17, 4, 2], we also use the notions of consistency and robustness to evaluate our algorithms. We denote the prediction error as $\zeta$, which is the absolute difference between the prediction and the actual outcome. We say that an online algorithm is $\alpha$-consistent if $\text{ALG} \leq \alpha \cdot \text{OPT}$ when the prediction is accurate, i.e., $\zeta = 0$, and $\beta$-robust if $\text{ALG} \leq \beta \cdot \text{OPT}$ for all $\zeta$ and feasible outcomes to the problem. We call $\alpha$ and $\beta$ the consistency factor and robustness factor, respectively. Thus consistency characterizes how well the algorithm does in case of perfect predictions, and robustness characterizes how well it does in worst-case predictions.

This novel analytical framework can bridge the gap between the aforementioned two radically online algorithm design methodologies, either pure worst case analysis or pure prediction based model. In this framework, a hyperparameter $\lambda \in (0, 1)$ is leveraged to determine the trust on ML predictions, where $\lambda = 0$ indicates fully trust on ML predictions and $\lambda = 1$ indicates no trust on ML predictions.

**Main Results.** Our main contribution is to develop online algorithms for MSSR with consistency and robustness properties in presence of ML predictions. We develop new analysis techniques for online algorithms with ML predictions via the hyperparameter. We first define a few notions before presenting our main results. We assume there are $n$ shops with buying prices $b_1 > \cdots > b_n$ and renting prices $r_1 < \cdots < r_n$. We develop several online algorithms for MSSR with prediction inputs from both a single ML algorithm and multiple ML algorithms as highlighted below:

• We first present a best deterministic algorithm (achieving minimal competitive ratio) for MSSR without ML predictions. It turns out that the algorithm chooses exactly one shop $i$ with the minimal value of $r_i + (b_i - r_i)/b_n$, and buy on the start day $b_n$ at shop $i$.

• Next, we consider MSSR with prediction from a single ML algorithm. We show that if this ML prediction is naively used in algorithm design, the proposed algorithm cannot ensure robustness (Section 3.1). We then incorporate ML prediction in a judicious manner by first proposing a deterministic online algorithm that is $((\lambda + 1)r_n + b_1/b_n)$-consistent, and $(\max\{r_n, b_1/b_n\} + 1/\lambda)$-robust (Section 3.2). We further propose a randomized algorithm with consistency and robustness guarantees (Section 3.3). We numerically evaluate the performance of our online algorithms (Section 3.4). We show that with a natural prediction error model, our algorithms are practical, and achieve better performance than the ones without ML predictions. We also investigate the impacts of several parameters and provide insights on the benefits of using ML predictions. It turns out that the predictions need to be carefully incorporated in online algorithm design.

• We then study a more general setting where we get predictions from $m$ ML algorithms. We propose both a deterministic algorithm (Section 4.1) and a randomized algorithm (Section 4.2) with consistency and robustness guarantees. Numerical results are given to demonstrate the impact of multiple ML predictions.

All detailed proofs in the paper are relegated to the supplemental material and can be found in [20].

**Related Work.** We borrow the concepts of consistency and robustness from [12], which incorporates ML predictions into the classical Marker algorithm ensuring both robustness and consistency for caching. The models have been extended for a comprehensive understanding of the classical ski rental problem [4, 17]. The impact of advice quality has been further quantified and a Pareto-optimal algorithm for ski rental problem has been proposed in [2]. While we operate in the same framework, none of previous results can be directly applied to our setting, as our work significantly differs from previous studies in the sense that we consider a multi-shop ski rental problem with predictions from multiple ML models, where the skier has to make a two-fold decision on when and where to buy. This makes the problem considerably more challenging but more practical.

Closest to our model is that multiple options in one shop [11] or multiple shops [1], however, no ML prediction is incorporated in their online algorithms design. On the other hand, there is an extensive study for online optimization with advice model, in particular, multiple predictions has been studied in the context of online learning, however, existing techniques are not applicable to our multi-shop setting. We refer interested readers to the surveys [3, 13] for a comprehensive discussion.

## 2 Preliminaries

We consider the *multi-shop ski rental* (MSSR) problem, where a skier goes to ski for an unknown number of days. The skier can *buy* or *rent* skis from multiple shops with different buying and renting prices. Specifically, we consider the case that *the skier must choose one shop as soon as she starts the skiing, and must rent or buy the skis at that particular shop since then In other words, once a shop is chosen by the skier, the only decision variable is when she should buy the skis.*

More precisely, we assume that there are totally $n$ shops and denote the set of shops as $\mathcal{N} = \{1, \cdots, n\}$. Each shop $i$ offers a renting price of $r_i$ dollars per day, and a buying price $b_i$ dollars, where $r_i, b_i > 0, \forall i \in \mathcal{N}$. In particular, our model reduces to the classical ski rental problem when $n = 1$. It is obvious that if one shop has higher prices for both renting and buying than another shop, it is *suboptimal* to choose this shop. To that end, we assume $0 < r_1 < \cdots < r_n$, and $b_1 > \cdots > b_n > 0$. For the ease of exposition, we set $r_1 = 1$, which is used in classical ski rental problem. Let $x$ be the actual number of skiing days which is *unknown* to the algorithm.

We first consider the offline optimal algorithm where $x$ is known. It is easy to see that the skier should rent at shop 1 if $x \le b_n$ and buy on day 1 at shop $n$ if $x > b_n$.

### 2.1 Best Deterministic Online Algorithm for MSSR

It is well-known that the best deterministic algorithm for the classical ski rental problem is the *break-even* algorithm: rent until day $b - 1$ and buy on day $b$. The corresponding CR is 2 and no other deterministic algorithm can do better. Now we consider the best deterministic online algorithm (BDOA) that obtains a minimal CR for MSSR without any ML prediction.

**Lemma 1.** *The best deterministic algorithm for MSSR is that the skier rents for the first $b_n - 1$ days and buys on day $b_n$ at shop $i$, where $i = \arg\min \left( r_i + \frac{b_i - r_i}{b_n} \right)$. The corresponding CR is $r_i + (b_i - r_i)/b_n$.*

**Remark 1.** *We consider a basic setting of MSSR, which occurs in many real-world applications. For example, in cloud computing systems, if the user decides to switch service from one cloud provider to another one, this might lead to a large amount of data transfer as well, which is very costly. Beyond this, there are several extensions of MSSR [1], e.g., (i) MSSR with* switching cost *(MSSR-S), i.e., the skier is able to switch from one shop to another at some non-zero costs; and (ii) MSSR with* entry fee *(MSSR-E), i.e., there is an entry fee for each shop and no switching is allowed. In MSSR/MSSR-E, the skier needs to decide* where *to rent or buy and* when *to buy the skis at the very beginning. In MSSR-S, the skier is able to decide where to rent or buy the skis* at any time. *MSSR-S and MSSR-E can be equivalently reduced to MSSR, e.g., switching happens only when buying [1]. Thus our proposed algorithms can be extended to these models with some minor changes in the constant terms.*

## 3 Online Algorithms for MSSR with Prediction from a Single ML Algorithm

In this section, we consider MSSR with prediction from a single ML algorithm. Let $y$ be the predicted number of skiing days. Then $\zeta = |y - x|$ is the prediction error. For the ease of exposition, we

use the two-shop ski rental problem as a motivating example, and then generalize the results to the general MSSR with $n$ shops.

---
**Algorithm 1** A simple learning-aided algorithm
---
    **if** $y \geq b_2$ **then**
        Buy on day 1 at shop 2
    **else**
        Rent at shop 1

---

### 3.1 A Simple Algorithm with ML prediction

**Lemma 2.** *The cost of Algorithm 1 satisfies ALG $\leq$ OPT $+ \zeta$.*

We now generalize Algorithm 1 and Lemma 2 to the MSSR with $n$ shops. Inspired by Lemma 1, it is easy to check that it is suboptimal to buy at shop $i$ with $b_i \geq b_n$, and rent at shop $j$ with $r_j > r_1$.

**Corollary 1.** *The simple algorithm with ML prediction for the general MSSR with $n$ shops follows that the skier buy on day 1 at shop $n$ if $y \geq b_n$, otherwise it rents at shop 1. The corresponding cost satisfies ALG $\leq$ OPT $+ \zeta$.*

**Remark 2.** *We note that by simply following the ML prediction, the CR of Algorithm 1 is unbounded (e.g., $x \gg b_2$) even when the prediction $y$ is small (due to case (iii) in the proof provided in supplemental material). Furthermore, Algorithm 1 has no robustness guarantee.*

In the following, we show how to properly integrate the ML prediction into online algorithm design to achieve both consistency and robustness.

### 3.2 A Deterministic Algorithm with Consistency and Robustness Guarantee

We develop a new deterministic algorithm by introducing a hyperparameter $\lambda \in (0,1)$, which gives us a smooth tradeoff between the consistency and robustness of the algorithm.

---
**Algorithm 2** A deterministic algorithm with consistency and robustness guarantee
---
    **if** $y \geq b_2$ **then**
        Rent until day $\lceil \lambda b_2 \rceil - 1$ at shop 2, then buy on day $\lceil \lambda b_2 \rceil$ at shop 2
    **else**
        Rent until day $\lceil \frac{b_1}{\lambda} \rceil - 1$ at shop 1, then buy on day $\lceil \frac{b_1}{\lambda} \rceil$ at shop 1

---

**Theorem 1.** *The CR of Algorithm 2 is at most* $\min\{(\lambda + 1)r_2 + \frac{b_1}{b_2} + \max\{\lambda r_2 + 1, \frac{b_1}{b_2} \cdot \frac{1}{1-\lambda}\}\frac{\zeta}{OPT}, \max\{r_2 + \frac{1}{\lambda}, \frac{b_1}{b_2}(1 + \frac{1}{\lambda})\}\}$, *where $\lambda \in (0,1)$ is a parameter. In particular, Algorithm 2 is $((\lambda + 1)r_2 + \frac{b_1}{b_2})$-consistent and $(\max\{r_2, \frac{b_1}{b_2}\} + \frac{1}{\lambda})$-robust.*

*Proof sketch of Theorem 1:* We provide the sketch of the proof. We first prove the first bound. When $y \geq b_2$, we consider three cases. (1) $x < \lceil \lambda b_2 \rceil$, OPT $= x$ and ALG $= r_2 x$, i.e., $CR_1 = r_2$. (2) $\lceil \lambda b_2 \rceil \leq x < b_2$, OPT $= x$, and ALG $= r_2(\lceil \lambda b_2 \rceil - 1) + b_2 \leq (\lambda r_2 + 1)b_2 \leq (\lambda r_2 + 1)y = (\lambda r_2 + 1)(x + \zeta) = (\lambda r_2 + 1)(OPT + \zeta)$, i.e., $CR_2 \leq (\lambda r_2 + 1)(1 + \zeta/OPT)$. (3) $x \geq b_2$, OPT $= b_2$, and ALG $= r_2(\lceil \lambda b_2 \rceil - 1) + b_2 \leq (\lambda r_2 + 1)b_2 \leq (\lambda r_2 + 1)(OPT + \zeta)$, the bound is same as $CR_2$. Similarly, when $y < b_2$, we consider three cases. (4) $x < b_2$, ALG $=$ OPT $= x$, i.e., CR $= 1$. (5) $x \in [b_2, \lceil b_1/\lambda \rceil)$, OPT $= b_2$, and ALG$=x=y + \zeta <$ OPT $+ \zeta$, i.e., $CR_3 < 1 + \zeta/OPT$. (6) $x \geq \lceil b_1/\lambda \rceil$, OPT $= b_2$, and ALG $= \lceil b_1/\lambda \rceil - 1 + b_1 \leq b_1/\lambda + b_1 < b_1 + \frac{b_1}{b_2}\frac{1}{1-\lambda}\zeta$, i.e., $CR_4 < \frac{b_1}{b_2}(1 + \frac{1}{1-\lambda}\frac{\zeta}{OPT})$. Combining $CR_1, CR_2, CR_3$ and $CR_4$, we get the first bound. Now we prove the second bound. When $y \geq b_2$, ALG $= r_2(\lceil \lambda b_2 \rceil - 1) + b_2$ if $x \geq \lceil \lambda b_2 \rceil$, and the worst CR is obtained when $x = \lceil \lambda b_2 \rceil$, for which OPT $= \lceil \lambda b_2 \rceil$. Therefore, ALG $\leq (\lambda r_2 + 1)b_2 \leq \frac{\lambda r_2 + 1}{\lambda}\lceil \lambda b_2 \rceil = (r_2 + 1/\lambda)$OPT. Similarly, when $y < b_2$, the worst CR is obtained when $x = \lceil b_1/\lambda \rceil$, for which OPT $= b_2$, and ALG $= \lceil b_1/\lambda \rceil - 1 + b_1 \leq b_1/\lambda + b_1 = \frac{b_1}{b_2}(1 + 1/\lambda)$OPT.

Similarly, we can generalize the above results to the general MSSR with $n$ shops.

**Corollary 2.** *The deterministic algorithm with a single ML prediction for the general MSSR with $n$ shops follows that the skier buys on day $\lceil \lambda b_n \rceil$ at shop $n$ if $y \geq b_n$, otherwise it buys on day $\lceil b_1/\lambda \rceil$ at shop 1. The corresponding CR is at most $\min\{(\lambda + 1)r_n + b_1/b_n + \max\{\lambda r_n + 1, \frac{b_1}{b_n}\frac{1}{1-\lambda}\}\frac{\zeta}{OPT}, \max\{r_n + \frac{1}{\lambda}, \frac{b_1}{b_n}(1+\frac{1}{\lambda})\}\}$, where $\lambda \in (0,1)$ is a parameter. In particular, the deterministic algorithm is $((\lambda + 1)r_n + b_1/b_n)$-consistent and $(\max\{r_n, b_1/b_n\} + 1/\lambda)$-robust.*

**Remark 3.** *The CR is a function of hyperparameter $\lambda$ and prediction error $\zeta$, which is different from the conventional competitive design. By tuning the $\lambda$ value, one can achieve different values for CR. The CR might be even worse than the BDOA for some cases (e.g., prediction error is large). We will show this in Section 3.4. This shows that decision making based on ML predictions comes at the cost of lower worst-case performance guarantee. Finally, it is possible to find the optimal $\lambda$ to minimize the worst-case CR if the prediction error $\zeta$ is known (e.g. from historically observed error values).*

**Remark 4.** *Algorithm 2 provides a way to tradeoff consistency and robustness. In particular, if the algorithm has a greater trust in the predictor, $\lambda$ will be set close to zero, which leads to a better CR if the error $\zeta$ is small. On the other hand, less trust in the predictor will set $\lambda$ close to one which will achieve a more robust algorithm.*

### 3.3 A Randomized Algorithm with Consistency and Robustness Guarantee

We consider a class of randomized algorithms for MSSR in this section. Similarly, we consider a hyperparameter $\lambda$ satisfying $\lambda \in (1/b_2, 1)$. First, we emphasize that a randomized algorithm that naively modifies the distribution used for randomized algorithm design for the classical ski rental algorithm (with or without predictions) fail to achieve a better consistency and robustness at the same time. We customize the distribution functions carefully by incorporating different renting and buying prices from different shops into the distributions, as summarized in Algorithm 3.

---

**Algorithm 3** A randomized algorithm with consistency and robustness guarantee

---

    **if** $y \geq b_2$ **then**
        Let $k = \lfloor \lambda b_2 \rfloor$
        Define $p_i = \left(\frac{b_2 - r_2}{b_2}\right)^{k-i} \cdot \frac{r_2}{b_2\left(1-(1-\frac{r_2}{b_2})^k\right)}$, for $i = 1, \cdots, k$
        Choose $j \in \{1, 2, ..., k\}$ randomly from the distribution defined by $p_i$
        Rent till day $j - 1$ at shop 2, then buy on day $j$ at shop 2
    **else**
        Let $l = \left\lceil \frac{b_1}{\lambda} \right\rceil$
        Define $q_i = \left(\frac{b_1 - 1}{b_1}\right)^{l-i} \cdot \frac{1}{b_1\left(1-(1-\frac{1}{b_1})^l\right)}$, for $i = 1, \cdots, l$
        Choose $j \in \{1, 2, ..., l\}$ randomly from the distribution defined by $q_i$
        Rent till day $j - 1$ at shop 1, then buy on day $j$ at shop 1

---

**Theorem 2.** *The CR of Algorithm 3 is at most $\min\left\{\frac{r_2\lambda}{1-e^{-r_2\lambda}}(1 + \frac{\zeta}{OPT}), \frac{b_1}{b_2}\max\left\{\frac{r_2}{1-e^{-r_2(\lambda-1/b_2)}}, \frac{1/\lambda+1/b_1}{1-e^{-1/\lambda}}\right\}\right\}$. In particular, Algorithm 3 is $\left(\frac{r_2\lambda}{1-e^{-r_2\lambda}}\right)$-consistent and $\left(\frac{b_1}{b_2}\max\left\{\frac{r_2}{1-e^{-r_2(\lambda-1/b_2)}}, \frac{1/\lambda+1/b_1}{1-e^{-1/\lambda}}\right\}\right)$-robust.*

*Proof sketch of Theorem 2:* We provide the sketch of the proof. We compute the CR of Algorithm 3 under four cases.

**Case 1.** $y \geq b_2$ and $x \geq k$. OPT $= \min\{b_2, x\}$. According to Algorithm 3, the cost is $(b_2 + (i-1)r_2)$. We have $\mathbb{E}[\text{ALG}] = \sum_{i=1}^{k}(b_2 + (i-1)r_2)p_i \leq \frac{r_2 k/b_2}{1-e^{-r_2 k/b_2}}b_2 \leq \frac{r_2\lambda}{1-e^{-r_2\lambda}}(\text{OPT} + \zeta)$.

**Case 2.** $y \geq b_2$ and $x < k$. OPT $= x$. If the skier buys the skis on day $i \leq x$, then it incurs a cost $(b_2 + (i-1)r_2)$, otherwise, the cost is $xr_2$. We have $\mathbb{E}[\text{ALG}] = \sum_{i=1}^{x}(b_2 + (i-1)r_2)p_i + \sum_{i=x+1}^{k}xr_2 p_i = \frac{r_2 x}{1-(1-\frac{r_2}{b_2})^k} \leq \frac{r_2}{1-e^{-r_2 k/b_2}}\text{OPT} \leq \frac{b_1}{b_2}\cdot\frac{r_2}{1-e^{-r_2(\lambda-1/b_2)}}\text{OPT}$. For consistency, we can rewrite the above inequality $\mathbb{E}[\text{ALG}] \leq \frac{r_2\cdot k/b_2}{1-e^{-r_2 k/b_2}}\text{OPT} + \frac{r_2\cdot\zeta/b_2}{1-e^{-r_2 k/b_2}}k \leq \frac{r_2\lambda}{1-e^{-r_2\lambda}}(\text{OPT} + \zeta)$.

***Case* 3.** $y < b_2$ and $x < l$. OPT $= \min\{b_2, x\}$. We have $\mathbb{E}[\text{ALG}] = \sum_{i=1}^{x}(b_1 + (i-1) \cdot 1)p_i + \sum_{i=x+1}^{l} x \cdot 1 \cdot p_i \leq \frac{x}{1-e^{-l/b_1}} \leq \frac{x}{1-e^{-1/\lambda}} \leq \frac{\lambda}{1-e^{-\lambda}}(\text{OPT}+\zeta) \leq \frac{r_2\lambda}{1-e^{-r_2\lambda}}(\text{OPT}+\zeta)$.

***Case* 4.** $y < b_2$ and $x \geq l$. OPT $= b_2$. We have $\mathbb{E}[\text{ALG}] \leq \frac{l}{1-e^{-l/b_1}} \leq \frac{b_1/b_2 \cdot (1/\lambda + 1/b_1)}{1-e^{-1/\lambda}}\text{OPT}$. We rewrite the above inequality to get consistency $\mathbb{E}[\text{ALG}] \leq \frac{1}{1-e^{-1/\lambda}}(\text{OPT}+\zeta) \leq \frac{r_2\lambda}{1-e^{-r_2\lambda}}(\text{OPT}+\zeta)$.

**Remark 5.** *According to Algorithm 3, for any particular value of $\lambda$, the day when the skis are bought is sampled based on two different probability distributions, which depend on the prediction received and rents until that day. Note that different from the classical randomized algorithm for ski rental, we customize the distribution functions carefully by incorporating different renting and buying prices from different shops into the distributions.*

Again, we can generalize Algorithm 3 to the general MSSR problem with $n$ shops. As it is suboptimal to rent at any shop besides shop 1 and buy at any shop besides shop $n$. The randomized algorithm for the general MSSR simply replaces shop 2 by shop $n$ with the corresponding $b_n$ and $r_n$ in Algorithm 3. Similarly, the corresponding competitive ratio can be achieved by replacing $b_2$ and $r_2$ in Theorem 2 by $b_n$ and $r_n$ of shop $n$. The hyperparameter should satisfy $\lambda \in (1/b_n, 1)$.

## 3.4 Model Validation and Insights

**Synthetic dataset.** We generate a synthetic dataset with $n = 6$ shops, the buying costs are $100, 95, 90, 85, 80, 75$ dollars with $b_1 = 100$ and $b_6 = 75$, and the renting costs $1, 1.05, 1.10, 1.15, 1.20, 1.25$ dollars with $r_1 = 1$ and $r_6 = 1.25$. Note that the actual values of $b_i$ and $r_i$ are not important as we can scale all these values by some constant factors. The actual number of skiing days $x$ is a random variable uniformly drawn from $[1, \Gamma]$, where $\Gamma < \infty$ is a constant. The predicted number of skiing days $y$ is set to $x + \epsilon$ where $\epsilon$ is drawn from a normal distribution with mean $\delta$ and standard variation $\sigma$. We vary either the value of $\sigma$ from 0 to $\Gamma$, or the value of $\delta$ to verify the consistency and robustness of our algorithms.

To characterize the impact of the hyperparameter $\lambda$ on the performance of our algorithms, we consider the values of $0.25, 0.5, 0.75$ and $1$ for $\lambda$. Note $\lambda = 1$ means that our algorithms ignore the ML prediction, and reduce to the algorithms without predictions. For each value of $\sigma$, we plot the average competitive ratio by running the corresponding algorithm over $10^4$ independent trials. We consider both unbiased and biased prediction errors in our experiments[1].

We first consider unbiased prediction errors, i.e., $\delta = 0$, to characterize the impact of $\Gamma$ and $\lambda$.

***The impact of* $\Gamma$.** As $x$ is uniformly drawn from $[1, \Gamma]$, $\Gamma$ is an important parameter that can impact the CR. We consider two possible values of $\Gamma$: $\Gamma = 3b_1$ and $\Gamma = b_1$. As $b_6 = 75$, $\Gamma = 3b_1$ means that it is highly possible the actual number of skiing days $x$ is larger than $b_6$. Thus according to Algorithm 2, buying as early as possible will be a better choice, i.e., small $\lambda$ results in better CR as shown in Figure 1 *(Left)*.

On the other hand, with $\Gamma = b_1$, it is highly possible that $x$ is smaller than $b_6$. Therefore, if

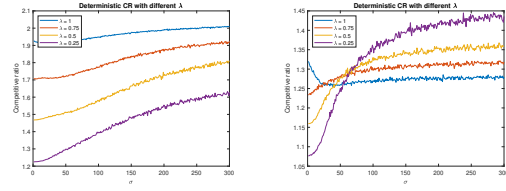

Figure 1: Impact of unbiased prediction errors on Algorithm 2. *(Left):*$\Gamma = 3b_1$; *(Right):*$\Gamma = b_1$.

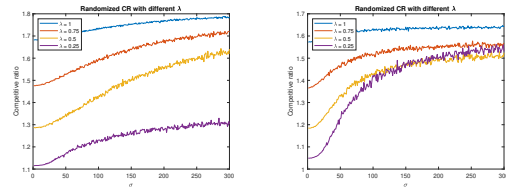

Figure 2: Impact of unbiased prediction errors on Algorithm 3. *(Left):*$\Gamma = 3b_1$; *(Right):*$\Gamma = b_1$.

the prediction is more accurate (small $\sigma$), smaller $\lambda$ (i.e., more trust on ML predictions) achieves smaller CR, while the prediction is inaccurate (with large $\sigma$), larger $\lambda$ achieves smaller CR. This can be observed from Figure 1 *(Right)*. In particular, with the values of $b$'s and $r$'s in our setting, $\lambda = 1$, i.e., do not trust the prediction achieves the best CR when the prediction error is large. We can observe a similar trend for the randomized algorithm (Algorithm 3) as shown in Figures 2.

***The impact of hyperparameter*** $\lambda$. We further compare the performance of the deterministic algorithm (Algorithm 2) and the randomized algorithms (Algorithm 3), as shown in Figure 3 with $\Gamma = 3b_1$. We make the following observations: (i) With the same prediction errors (e.g., $\lambda = 0.5$), the randomized algorithm always performs better than the deterministic algorithm. Similar trends are observed for other $\lambda$ values and hence are omitted due to space constraints. (ii) Our deterministic algorithm with ML prediction can beat the performance of classical randomized algorithm without ML predictions when the standard deviation of prediction error is smaller than $2.5b_1 = 250$.

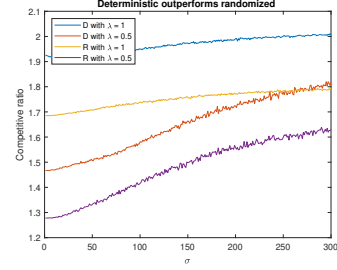

Figure 3: CR of Algorithm 2 vs. Algorithm 3.

Hyperparameter $\lambda$ incorporates the trust of ML predictions in online algorithm design. In particular, $\lambda$ close to $0$ means more trust on predictions while $\lambda$ close to $1$ means less trust. We investigate its impact on Algorithm 2 by considering a perfect prediction and an extremely erroneous prediction. From Figure 4 with $\Gamma = 3b_1$, we observe (i) With an extremely erroneous prediction, blinding trust the prediction (smaller $\lambda$) leads to worse performance than BDOA without ML predictions. (ii) By properly choosing $\lambda$, our algorithm achieves better performance than BDOA even with extremely erroneous prediction. This demonstrates the importance of hyperparameter $\lambda$.

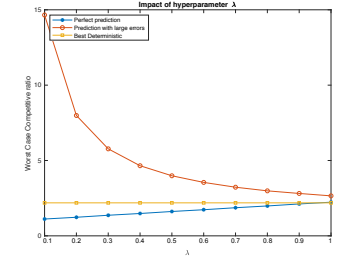

Figure 4: Impact of hyperparameter.

Next we consider the impact of biases on prediction errors. We consider three possible values of $10, 20, 50$ for $\delta$. The performance of Algorithm 2, and Algorithm 3 with $\Gamma = 3b_1$ are shown in Figure 5. With the above analysis of $\Gamma$'s impact and the same trust on ML predictions ($\lambda = 0.5$), a smaller bias benefits the CR when the variance is small, however, when the variance is large, the impact of bias is negligible. Similar trends are observed for other values of $\lambda$ and with $\Gamma = b_1$ and hence are relegated to the supplemental material.

**Real-world dataset.** We consider the viewer information for *The Big Bang Theory* (season 12), which consists of $24$ episodes [19]. Viewers can either buy the whole season at once or purchase each episode one by one, which corresponds to "buy" or "rent" in MSSR. There are two shops, *Google Play* and *Amazon Prime Video*, for viewers to choose with different buying and renting prices. Google (Amazon) offers a buying price of \$29.99 (\$19.99) and a renting price of \$1.99

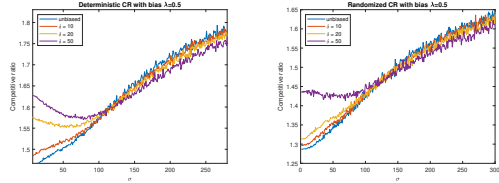

Figure 5: Impact of biased errors with $\Gamma = 3b_1$. *(Left):* Algorithm 2; *(Right):* Algorithm 3.

(\$2.99). Given the total viewers for each episode, we generate a probability distribution on the actual number of episodes watched by a viewer.

To characterize the impact of hyperparameter, we generate three models to predict the number of episodes watched by a new viewer. First, we generate a similar distribution on the number of episodes watched by viewers for the season $11$, and randomly draw the prediction $y$ from that distribution. We call this "Prediction 1". We then generate two other ("bad") predictions where "Prediction 2" follows that $y = 24 - x$, and "Prediction 3" satisfies $y = 1$ if $x \geq b_2$ and $y = 24$ otherwise.

Some notable observations from Figure 6 are: (i) With perfect prediction and $\lambda = 0$, our algorithm achieves the optimal performance, i.e., CR $= 1$. (ii) Improper values of $\lambda$ that leads to high trust on prediction will lead to even worse performance than pure online algorithm. For example, for "Prediction 2" with $\lambda < 0.2$, and "Prediction 3" with $\lambda < 0.4$. (iii) With proper value of $\lambda$, our algorithm achieves better performance than pure online algorithm even with erroneous predictions. For example, for "Prediction 1" with $0 < \lambda < 0.8$, "Prediction

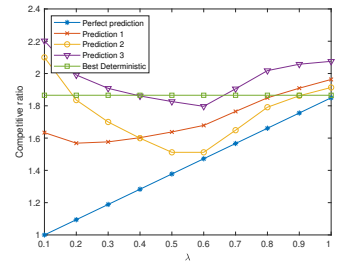

Figure 6: CR of Algorithm 2 with real-world dataset.

2" with $0.2 < \lambda < 0.9$ and "Prediction 3" with $0.4 < \lambda < 0.65$. This further demonstrate the importance of setting right values for the hyperparameter. *More importantly, we conclude that online algorithms with ML advice cannot always outperform pure online algorithms regardless of the values for the hyperparameter. However, it is always possible to find the right hyperparameter value such that the performance of online algorithm with ML advice is better than pure online ones.*

# 4 Online Algorithms with Prediction from Multiple ML Algorithms

Now we consider a more general case with predictions from $m$ ML algorithms, and denote them as $y_1, \cdots, y_m$. Without loss of generality, we assume $y_1 < y_2 < \cdots < y_m$. We define an indicator function $f(i)$ to represent the relation between $y_i$ and $b_n$, satisfying $f(i) = 1$ if $y_i \geq b_n$ and $f(i) = 0$ otherwise. Let $z = \sum_{i=1}^{m} f(i)$, which indicates the number of predictions that are greater than $b_n$. We redefine the prediction error under the multiple predictions case as $\zeta = \max_i |y_i - x|$.

We design both deterministic and randomized algorithms for MSSR with multiple ML predictions. Rather than comparing a single prediction with a threshold as in Section 3, we now determine whether the "majority" of the predictions are beyond the threshold and use this information to decide the "break-even point". Again for the ease of exposition, we take the two-shop ski rental problem as a motivating example, and the results can be easily generalized to the general $n$-shop MSSR.

## 4.1 A Deterministic Algorithm with Consistency and Robustness Guarantee

We first design a deterministic algorithm tuned by a hyperparameter $\lambda \in (0, 1)$ to achieve a tradeoff between consistency and robustness.

---
**Algorithm 4** A deterministic algorithm with multiple ML predictions

---
    **if** $z \geq m/2$ **then**

        Rent at shop 2 until buying on day $\left\lceil \frac{\lambda b_2}{2z-m+1} \right\rceil$ at shop 2

    **else**

        Rent at shop 1 until buying on day $\left\lceil \frac{(m-2z+1)b_2}{\lambda} \right\rceil$ at shop 1

---

**Theorem 3.** *The CR of Algorithm 4 is at most* $\min\{(\lambda + 1)r_2 + \frac{b_1}{b_2} + \max\{\lambda r_2 + 1, \frac{1}{1-\lambda}\}\frac{\zeta}{OPT}, \max\{r_2, \frac{b_1}{b_2}\} + \frac{m+1}{\lambda}\}$, *where* $\lambda \in (0, 1)$ *is a parameter. In particular, Algorithm 4 is* $\left((\lambda + 1)r_2 + \frac{b_1}{b_2}\right)$*-consistent and* $\left(\max\{r_2 + \frac{b_1}{b_2}\} + \frac{m+1}{\lambda}\right)$*-robust.*

**Remark 6.** *We add a term $+1$ into the break-even point in Algorithm 4 as $2z - m$ or $m - 2z$ may equal $0$ when $z$ is an even number. We numerically evaluate its impact in Section 4.3.*

## 4.2 A Randomized Algorithm with Consistency and Robustness Guarantee

In this section, we propose a randomized algorithm with multiple ML predictions that achieves a better tradeoff between consistency and robustness than the deterministic algorithm.

**Theorem 4.** *The competitive ratio of Algorithm 5 is at most* $\min\left\{\frac{b_1}{b_2}\max\left\{\frac{r_2}{1-e^{-r_2(\lambda/(m+1)-1/b_2)}},\right.\right.$ $\left.\left.\frac{m+1/\lambda+1/b_1}{1-e^{-1/\lambda}}\right\}, \frac{r_2\lambda}{1-e^{-r_2\lambda/(m+1)}}\left(1 + \frac{\zeta}{OPT}\right)\right\}$. *In particular, Algorithm 5 is* $\left(\frac{r_2\lambda}{1-e^{-r_2\lambda/(m+1)}}\right)$*-consistent and* $\left(\frac{b_1}{b_2}\max\left\{\frac{r_2}{1-e^{-r_2(\lambda/(m+1)-1/b_2)}}, \frac{m+1/\lambda+1/b_1}{1-e^{-1/\lambda}}\right\}\right)$*-robust.*

## 4.3 Model Validation and Insights

We consider the same synthetic setting as that in Section 3.4. We vary the number of ML predictions from 1 to 8, and set the associated predictions to $x + \epsilon$, where $\epsilon$ is drawn from a normal distribution with mean $\delta$ and standard variation $\sigma$, and $\Gamma = b_1$. We investigate the impacts of $m, \lambda$ and $\delta$ on the performance and make the following observations:

**Algorithm 5** A randomized algorithm with multiple ML predictions

---

**if** $z \geq m/2$ **then**

    Let $k = \left\lfloor \frac{\lambda b_2}{2z - m + 1} \right\rfloor$

    Define $p_i = \left( \frac{b_2 - r_2}{b_2} \right)^{k-i} \cdot \frac{r_2}{b_2 \left( 1 - (1 - \frac{r_2}{b_2})^k \right)}$, for $i = 1, \cdots, k$

    Choose $j \in \{1, 2, ..., k\}$ randomly from the distribution defined by $p_i$

    Rent till day $j - 1$ and then buy on day $j$ at shop 2

**else**

    Let $l = \left\lceil \frac{m - 2z + 1}{\lambda} b_1 \right\rceil$

    Define $q_i = \left( \frac{b_1 - 1}{b_1} \right)^{l-i} \cdot \frac{1}{b_1 \left( 1 - (1 - \frac{1}{b_1})^l \right)}$, for $i = 1, \cdots, l$

    Choose $j \in \{1, 2, ..., l\}$ randomly from the distribution defined by $q_i$

    Rent till day $j - 1$ and then buy on day $j$ at shop 1

---

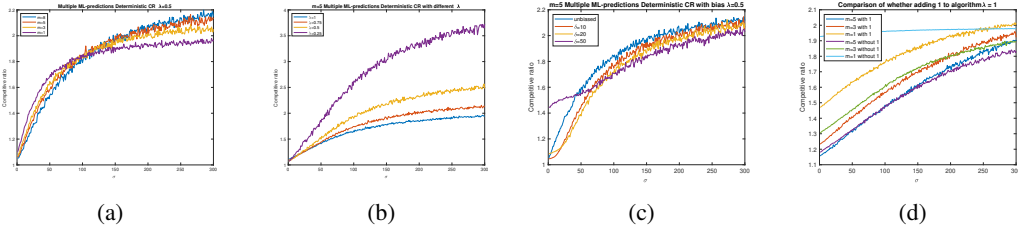

|  (a)  |  (b)  |  (c)  |  (d)  |

Figure 7: CR of Algorithm 4 under unbiased errors for (a) $\lambda = 0.5$ with different $m$ and (b) $m = 5$ with different $\lambda$. (c) CR of Algorithm 4, $m = 5$ and $\lambda = 0.5$ under biased errors. (d) The impact of term $+1$ in the deterministic algorithm design.

▷ (i) For unbiased prediction errors and fixed $\lambda$, if the prediction is accurate (small $\sigma$), increasing $m$ improves the competitive ratio, however, more predictions hurt the competitive ratio when prediction error is large, see Figure 7 (a).

▷ (ii) For $\delta = 0$ with fixed $m = 5$, if the prediction is accurate, more trust (small $\lambda$) benefits the algorithm. On the other hand, less trust achieves better competitive ratio when the prediction error is large. See Figure 7 (b).

▷ (iii) For fixed $m$ and $\lambda$, a smaller bias benefits the competitive ratio when the variance is small, while a larger bias achieves a smaller competitive ratio when the variance is large. See Figure 7 (c).

▷ (iv) We also characterize the impact of the "$+1$" term in Algorithm 4, and compare the algorithms with "$+1$" and without it in the break-even points, see Figure 7 (d). We observe that the "$+1$" can improve the competitive ratio as it suggests the skier to buy earlier when more predictions are above $b_2$, and rent longer when more predictions are smaller than $b_2$, i.e., making decisions more cautious. Similar trends can be observed when using other parameter values and using real-world dataset and hence we relegate the results to the supplemental material.

## 5   Conclusions

In this paper, we investigate how to improve the worst-case performance of online algorithms with predictions from (multiple) ML algorithms. In particular, we consider the general multi-shop ski rental problem. We develop both deterministic and randomized algorithms. Our online algorithms achieve a smooth tradeoff between consistency and robustness, and can significantly outperform the ones without ML predictions. Going further, we will study extensions of MSSR. e.g., the skier is allowed to switch shops, in which she can simultaneously decide where to buy or rent the skis. We will also consider to integrate prediction costs into the online algorithm design.

## Broader Impact

Dealing with *uncertainty* has been one of the most challenging issues that real-world application faces. Two radically different design methodologies for online decision making have been studied to deal with the uncertainty of future inputs. On the one hand, the competitive analysis framework has been widely used that "pessimistically" assumes that the future inputs are *unpredictable* and are always the *worst-case*. The goal here is to design online algorithms with a bounded *competitive ratio* in the worst-case over all feasible inputs. However, competitive algorithms are usually *conservative* and do not do well in the typical scenarios encountered in practice that are far from the worst-case. On the other hand, online algorithms implemented in real systems seldom assume the worst-case future inputs. Rather, they often use historical data to make predictions and use them as advice in decision making. These "optimistic" algorithms work well if the future inputs look similar to past ones and may perform poorly when these assumptions are violated.

The framework proposed in this paper bridges the gap between the two extreme worlds of pessimistic and optimistic algorithm design by incorporating machine learned advices from machine learning models. Given that online decision making with uncertainty is at the core of our daily life, the scientific knowledge and tools developed from our work will advance the state-of-the-art methods and take a significant stride toward bringing benefits and better experiences to users, service providers and society at large.

As (online) algorithms will continue to spread everywhere with both visible and invisible benefits, it also arises some concerns. For example, human judgement might be lost when data and predictive modeling become paramount. Furthermore, there are biases in algorithmically organized systems since algorithms depend on data and reflect the biases of datasets. Further studies on online algorithms design with ML advice to address these issues will be interesting.

## Acknowledgments and Disclosure of Funding

This research of Shiqiang Wang was sponsored by the U.S. Army Research Laboratory and the U.K. Ministry of Defence under Agreement Number W911NF-16-3-0001. The views and conclusions contained in this document are those of the authors and should not be interpreted as representing the official policies, either expressed or implied, of the U.S. Army Research Laboratory, the U.S. Government, the U.K. Ministry of Defence or the U.K. Government. The U.S. and U.K. Governments are authorized to reproduce and distribute reprints for Government purposes notwithstanding any copyright notation hereon.

## Footnotes

[1]The source code of our simulation is available at https://github.com/ShufanWangBGM/OAfMSSRwMLA.

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
