[Supplementary Material]

# Online Algorithms for Multi-shop Ski Rental with Machine Learned Advice Supplementary Material

**Shufan Wang**
Binghamton University
State University of New York
Binghamton, NY 13902
swang214@binghamton.edu

**Jian Li**
Binghamton University
State University of New York
Binghamton, NY 13902
lij@binghamton.edu

**Shiqiang Wang**
IBM Thomas J. Watson Research Center
Yorktown Heights, NY 10598
wangshiq@us.ibm.com

## 1 Examples of Real-World Applications

Here we give a few real world applications that can be modeled with MSSR.

**Example 1: Cost in Cloud CDN Service.** With the advent of cloud computing, the content service provided by content distribution network (CDN) has been offered as managed platforms with a novel pay-as-you-go model for cloud CDNs. For example, cloud providers such as Microsoft Azure and Amazon AWS, now provide different price options to users based on their demand, which is usually unknown in advance. Table 1 lists the price option provided by Microsoft Azure. Each price option can be considered as a shop in the MSSR problem, and the hourly price is the renting price.

| Options | Hourly price ($) |
|---|---|
| Pay-as-you-go | 0.0075 |
| 1 year reserved | 0.0059 |
| 3 year reserved | 0.0038 |

Table 1: Price option for Microsoft Azure basic service.

**Example 2: Caching.** A content can be replicated and stored in multiple base stations to serve requests from users. Upon a user request, if the requested content is stored in base stations, the service latency is short, otherwise, it incurs a longer latency to fetch the requested content from remote servers. On the other hand, the content can be prefetched and stored in base stations at the expense of wasting space if the content will not be requested by users. In this application, each base station is considered as a shop, and renting corresponds to serve requests on-demand, and buying refers to prefetch content in advance.

## 2 Proof of Lemma 1

It is obvious that $\text{OPT} = \min\{x, b_n\}$. Since the skier cannot change the shop once she chooses it under our model, we can consider the competitive ratio of shop $\forall i \in \mathcal{N}$. Let $d_i$ be the buying day. Then $\text{ALG}_i = xr_i$ if $x < d_i$, otherwise $\text{ALG}_i = (d_i - 1)r_i + b_i$. It is easy to argue that the worst

case happens when $x = d_i$. We have

$$\begin{aligned}
\mathrm{CR}_i &= \frac{\mathrm{ALG}_i}{\mathrm{OPT}} = \frac{(d_i - 1)r_i + b_i}{\min\{x, b_n\}} = \frac{(d_i - 1)r_i + b_i}{\min\{d_i, b_n\}} \\
&= \frac{(d_i + b_n)r_i + b_i - r_i - b_n r_n}{\min\{d_i, b_n\}} \\
&= \frac{(\min\{d_i, b_n\} + \max\{d_i, b_n\})r_i + b_i - r_i - b_n r_n}{\min\{d_i, b_n\}} \\
&= r_i + \frac{\max\{d_i, b_n\}r_i + b_i - r_i - b_n r_n}{\min\{d_i, b_n\}}.
\end{aligned}$$

Hence, the competitive ratio is minimized when $d_i = b_n$, i.e., the best competitive ratio satisfies $\mathrm{CR}_i = r_i + (b_i - r_i)/b_n$. Thus, we have $\mathrm{CR} = \min_i \mathrm{CR}_i$.

## 3   Proof of Lemma 2

Since there is only one break-even point $b_2$, we consider four cases based on the relations of $x$ and $y$ with $b_2$.

(i) $y \geq b_2$ and $x \geq b_2$: $\mathrm{ALG} = b_2$, $\mathrm{OPT} = b_2$, i.e., $\mathrm{CR} = 1$;

(ii) $y \geq b_2$ and $x < b_2$: $\mathrm{ALG} = b_2$, $\mathrm{OPT} = x$, i.e., $\mathrm{CR} = b_2/x$;

(iii) $y < b_2$ and $x \geq b_2$: $\mathrm{ALG} = x$, $\mathrm{OPT} = b_2$, i.e., $\mathrm{CR} = x/b_2$;

(iv) $y < b_2$ and $x < b_2$: $\mathrm{ALG} = x$, $\mathrm{OPT} = x$, i.e., $\mathrm{CR} = 1$.

Combining (i)-(iv), $\mathrm{CR} = \max\{b_2/x, x/b_2\}$, which is unbounded.

Furthermore, we can rewrite (ii), $\mathrm{ALG} = b_2 = x + b_2 - x \leq \mathrm{OPT} + y - x = \mathrm{OPT} + \zeta$.

Similarly, by rewriting (iv), we also have $\mathrm{ALG} = x = b_2 + x - b_2 < \mathrm{OPT} + x - y = \mathrm{OPT} + \zeta$.

## 4   Proof of Theorem 1

We first prove the first bound. When $y \geq b_2$, we consider two cases.

First, if $x < \lceil \lambda b_2 \rceil$, then $\mathrm{OPT} = x$, i.e., rent at shop 1 since $r_1 = 1 < r_2$. Hence we have

$$\mathrm{ALG} = r_2 x = r_2 \mathrm{OPT},$$

i.e., $\mathrm{CR}_1 = r_2$.

Second, if $x \geq \lceil \lambda b_2 \rceil$, we have

$$\mathrm{ALG} = r_2(\lceil \lambda b_2 \rceil - 1) + b_2 \leq (\lambda r_2 + 1)b_2.$$

When $x \geq b_2$, we have $\mathrm{OPT} = b_2$, i.e., buy at shop 2 on day 1 as $b_2 < b_1$, then

$$\mathrm{ALG} \leq (\lambda r_2 + 1)b_2 \leq (\lambda r_2 + 1)(\mathrm{OPT} + \zeta).$$

When $\lceil \lambda b_2 \rceil \leq x < b_2$, we have $\mathrm{OPT} = x$, then $b_2 \leq y = x + \zeta = \mathrm{OPT} + \zeta$, thus,

$$\mathrm{ALG} \leq (\lambda r_2 + 1)b_2 \leq (\lambda r_2 + 1)(\mathrm{OPT} + \zeta).$$

Combining these two cases, we have $\mathrm{CR}_2 \leq (\lambda r_2 + 1)(1 + \frac{\zeta}{\mathrm{OPT}})$.

Similarly, when $y < b_2$, we consider the following three cases.

First, if $x < b_2$, we have $\mathrm{ALG} = x$. It is clear that $\mathrm{OPT} = x$, i.e., $\mathrm{CR} = 1$.

Second, if $x \in \left[b_2, \lceil \frac{b_1}{\lambda} \rceil\right)$, we have $\mathrm{OPT} = b_2$, i.e., buy at shop 2 on day 1, and

$$\mathrm{ALG} \overset{(a)}{=} x \overset{(b)}{=} y + \zeta < \mathrm{OPT} + \zeta,$$

where (a) is obtained by following Algorithm 2, i.e., rent at shop 1 with $r_1 = 1$, and (b) holds true due to the predictor error definition. Therefore, we have $\mathrm{CR}_3 < 1 + \frac{\zeta}{\mathrm{OPT}}$.

Finally, if $x \geq \lceil \frac{b_1}{\lambda} \rceil$, we have OPT $= b_2$, and

$$\text{ALG} = \left\lceil \frac{b_1}{\lambda} \right\rceil - 1 + b_1 \leq \frac{b_1}{\lambda} + b_1 \overset{(c)}{<} b_1 + \frac{b_1}{b_2} \frac{1}{1-\lambda} \zeta,$$

where (c) follows $\zeta = x - y > \frac{b_2}{\lambda} - b_2$, i.e., $b_2 < \frac{\lambda}{1-\lambda} \zeta$, then $b_1 < \frac{b_1}{b_2} \frac{\lambda}{1-\lambda} \zeta$. Thus $\text{CR}_4 < \frac{b_1}{b_2}(1 + \frac{1}{1-\lambda} \frac{\zeta}{\text{OPT}})$.

Combining $\text{CR}_1, \text{CR}_2, \text{CR}_3$ and $\text{CR}_4$, we get the first bound.

Now we prove the second bound. According to Algorithm 2, the skier rents the skis at shop 2 until day $\lceil \lambda b_2 \rceil - 1$ and then buys on day $\lceil \lambda b_2 \rceil$ at shop 2, when the predicted day satisfies $y \geq b_2$, we have

$$\text{ALG} = r_2(\lceil \lambda b_2 \rceil - 1) + b_2,$$

if $x \geq \lceil \lambda b_2 \rceil$. It is easy to see that the worst CR is obtained when $x = \lceil \lambda b_2 \rceil$, for which OPT $= \lceil \lambda b_2 \rceil$. Therefore,

$$\text{ALG} \leq (\lambda r_2 + 1)b_2 \leq \frac{\lambda r_2 + 1}{\lambda} \lceil \lambda b_2 \rceil = \left( r_2 + \frac{1}{\lambda} \right) \text{OPT}.$$

Similarly, the skier rents the skis at shop 1 until day $\lceil \frac{b_1}{\lambda} \rceil - 1$ and then buys on day $\lceil \frac{b_1}{\lambda} \rceil$ at shop 1, when $y < b_2$, the worst CR is obtained when $x = \lceil \frac{b_1}{\lambda} \rceil$, for which OPT $= b_2$, and

$$\text{ALG} = \left\lceil \frac{b_1}{\lambda} \right\rceil - 1 + b_1 \leq \frac{b_1}{\lambda} + b_1 = \frac{b_1}{b_2} \left( 1 + \frac{1}{\lambda} \right) \text{OPT}.$$

## 5 Proof of Theorem 2

We compute the competitive ratio of Algorithm 3 under four cases.

**Case 1.** $y \geq b_2$ and $x \geq k$. It is clear that OPT $= \min\{b_2, x\}$. According to Algorithm 3, the skier should rent at shop 2 until day $j - 1$ and buy on day $j$. This happens with probability $p_i$, for $i = 1, \cdots, k$, and incurs a cost $(b_2 + (i-1)r_2)$. Therefore, we have Therefore, we have

$$
\begin{aligned}
\mathbb{E}[\text{ALG}] &= \sum_{i=1}^{k} (b_2 + (i-1)r_2)p_i \\
&= \sum_{i=1}^{k} (b_2 + (i-1)r_2) \left( \frac{b_2 - r_2}{b_2} \right)^{k-i} \cdot \frac{r_2}{b_2 \left( 1 - (1 - \frac{r_2}{b_2})^k \right)} \\
&= \frac{r_2 k}{1 - (1 - \frac{r_2}{b_2})^k} \overset{(a)}{\leq} \frac{r_2 k / b_2}{1 - e^{-r_2 k/b_2}} b_2 \overset{(b)}{\leq} \frac{r_2 \lambda}{1 - e^{-r_2 \lambda}} (\text{OPT} + \zeta),
\end{aligned}
$$

where (a) holds since $(1 + x)^k \leq e^{kx}$, for $0 \leq x < 1$, and (b) follows that $k = \lfloor \lambda b_2 \rfloor \leq \lambda b_2$, i.e., $k/b_2 \leq \lambda$ and $\frac{x}{1 - e^{-x}}$ increases in $x \geq 0$.

**Case 2.** $y \geq b_2$ and $x < k$. Since $x < k = \lfloor \lambda b_2 \rfloor < b_2$, we have OPT $= x$. If the skier buys the skis on day $i \leq x$, then it incurs a cost $(b_2 + (i-1)r_2)$, otherwise, the cost is $xr_2$. Therefore, we obtain the robustness through the following

$$
\begin{aligned}
\mathbb{E}[\text{ALG}] &= \sum_{i=1}^{x} (b_2 + (i-1)r_2)p_i + \sum_{i=x+1}^{k} xr_2 p_i \\
&= \frac{r_2}{b_2 \left( 1 - (1 - \frac{r_2}{b_2})^k \right)} \left[ \sum_{i=1}^{x} (b_2 + (i-1)r_2) \left( \frac{b_2 - r_2}{b_2} \right)^{k-i} + \sum_{i=x+1}^{k} xr_2 \left( \frac{b_2 - r_2}{b_2} \right)^{k-i} \right] \\
&= \frac{r_2 x}{1 - (1 - \frac{r_2}{b_2})^k} \leq \frac{r_2}{1 - e^{-r_2 k/b_2}} \text{OPT} \overset{(c)}{\leq} \frac{b_1}{b_2} \cdot \frac{r_2}{1 - e^{-r_2(\lambda - 1/b_2)}} \text{OPT},
\end{aligned}
$$

where (c) holds true since $\lambda b_2 - 1 \leq k = \lfloor \lambda b_2 \rfloor < b_2$, i.e., $k/b_2 \geq \lambda - 1/b_2$, and $b_1/b_2 > 1$. To get the consistency, we can rewrite the above inequality

$$
\mathbb{E}[\text{ALG}] \leq \frac{r_2}{1 - e^{-r_2 k/b_2}} \text{OPT}
$$

$$
\overset{(d)}{=} \frac{r_2 \cdot k/b_2}{1 - e^{-r_2 k/b_2}} \text{OPT} + \frac{r_2 \cdot (b_2 - k)/b_2}{1 - e^{-r_2 k/b_2}} x
$$

$$
\overset{(e)}{\leq} \frac{r_2 \cdot k/b_2}{1 - e^{-r_2 k/b_2}} \text{OPT} + \frac{r_2 \cdot \zeta/b_2}{1 - e^{-r_2 k/b_2}} k
$$

$$
= \frac{r_2 \cdot k/b_2}{1 - e^{-r_2 k/b_2}} \text{OPT} + \frac{r_2 \cdot k/b_2}{1 - e^{-r_2 k/b_2}} \zeta
$$

$$
\overset{(f)}{\leq} \frac{r_2 \lambda}{1 - e^{-r_2 \lambda}} (\text{OPT} + \zeta),
$$

where (d) follows $\text{OPT} = x$, (e) holds true since $x < k$, $y \geq b_2$, and $\zeta = y - x \geq b_2 - k$, and (f) follows that $k/b_2 \leq \lambda$.

***Case* 3.** $y < b_2$ and $x < l$. It is clear that $\text{OPT} = \min\{b_2, x\}$. Similar to *Case 2*, we have

$$
\mathbb{E}[\text{ALG}] = \sum_{i=1}^{x} (b_1 + (i-1) \cdot 1) p_i + \sum_{i=x+1}^{l} x \cdot 1 \cdot p_i = \frac{x}{1 - (1 - 1/b_1)^l}
$$

$$
\leq \frac{x}{1 - e^{-l/b_1}} \overset{(g)}{\leq} \frac{x}{1 - e^{-1/\lambda}} \overset{(h)}{\leq} \frac{\lambda}{1 - e^{-\lambda}} (\text{OPT} + \zeta) \overset{(i)}{\leq} \frac{r_2 \lambda}{1 - e^{-r_2 \lambda}} (\text{OPT} + \zeta),
$$

where (g) follows that $l = \lceil b_1/\lambda \rceil \geq b_1/\lambda$, i.e., $1/\lambda \leq l/b_1$, (h) follows from two cases i) when $x < b_2$, we have $\text{OPT} = x \geq x - \zeta$; and ii) when $x \geq b_2$, we have $x < x + b_2 - y = b_2 + \zeta$ as $y < b_2$, thus $b_2 > x - \zeta$. Hence, $\text{OPT} = b_2 \geq x - \zeta$. (i) holds since $r_2 > 1$ and $\frac{x}{1 - e^{-x}}$ increases in $x \geq 0$ as mentioned earlier.

***Case* 4.** $y < b_2$ and $x \geq l$. As $x \geq l$, we have $\text{OPT} = b_2$. Similar to *Case 1*, we have the robustness as

$$
\mathbb{E}[\text{ALG}] = \sum_{i=1}^{l} (b_1 + (i-1) \cdot 1) p_i = \frac{l}{1 - (1 - 1/b_1)^l} \leq \frac{l}{1 - e^{-l/b_1}}
$$

$$
= \frac{\lceil b_1/\lambda \rceil}{1 - e^{-l/b_1}} \overset{(j)}{\leq} \frac{b_2 \cdot \frac{b_1}{b_2} (\frac{1}{\lambda} + \frac{1}{b_1})}{1 - e^{-1/\lambda}} = \frac{\frac{b_1}{b_2} (\frac{1}{\lambda} + \frac{1}{b_1})}{1 - e^{-1/\lambda}} \text{OPT},
$$

where (j) follows that $\lceil b_1/\lambda \rceil \leq b_1/\lambda + 1 = b_1(1/\lambda + 1/b_1)$, and $l = \lceil b_1/\lambda \rceil \geq b_1/\lambda$, i.e., $l/b_1 \geq 1/\lambda$. Again, we rewrite the above inequality to get the consistency

$$
\mathbb{E}[\text{ALG}] \leq \frac{l}{1 - e^{-l/b_1}} \leq \frac{l}{1 - e^{-1/\lambda}} = \frac{b_2 + l - b_2}{1 - e^{-1/\lambda}} \overset{(k)}{\leq} \frac{1}{1 - e^{-1/\lambda}} (\text{OPT} + \zeta)
$$

$$
\leq \frac{\lambda}{1 - e^{-\lambda}} (\text{OPT} + \zeta) \leq \frac{r_2 \lambda}{1 - e^{-r_2 \lambda}} (\text{OPT} + \zeta),
$$

where (k) follows that $\text{OPT} = b_2$ and $\zeta = x - y > l - b_2$.

# 6 Proof of Theorem 3

We first prove the first bound. When $z \geq m/2$, we consider two cases.

First, if $x < \left\lceil \frac{\lambda b_2}{2z - m + 1} \right\rceil$, then $\text{OPT} = x$, i.e., rent at shop 1 since $r_1 = 1 < r_2$. Hence,

$$
\text{ALG} = r_2 x = r_2 \text{OPT},
$$

i.e., $\text{CR}_1 = r_2$.

Second, if $x \geq \left\lceil \frac{\lambda b_2}{2z - m + 1} \right\rceil$, then $\text{OPT} = \min\{b_2, x\}$ and

$$
\text{ALG} = r_2 \left( \left\lceil \frac{\lambda b_2}{2z - m + 1} \right\rceil - 1 \right) + b_2 \leq \left( \frac{\lambda}{2z - m + 1} r_2 + 1 \right) b_2 \overset{(a)}{\leq} (\lambda r_2 + 1) (\text{OPT} + \zeta),
$$

where (a) follows from two cases (i) when $\left\lceil \frac{\lambda b_2}{2z-m+1} \right\rceil \le x < b_2$, we have OPT $= x$, and $\zeta \ge y_m - x > b_2 - x$, i.e., $b_2 \le$ OPT $+ \zeta$; (ii) $x \ge b_2$, we have OPT $= b_2$, then $b_2 \le$ OPT $+ \zeta$. Furthermore, we have $2z - m + 1 \ge 1$. Hence $CR_2 = (\lambda r_2 + 1)\left(1 + \frac{\zeta}{\text{OPT}}\right)$.

Similarly, when $z < m/2$, we consider the following three cases.

First, if $x < b_2$, we have ALG $= x$. It is clear that OPT $= x$, i.e., rent at shop 1. Therefore, we have CR $= 1$.

Second, if $x \in \left[ b_2, \left\lceil \frac{(m-2z+1)b_2}{\lambda} \right\rceil \right)$, we have OPT $= b_2$, i.e., buy at shop 2 on day 1, and

$$\text{ALG} \overset{(b)}{=} x \overset{(c)}{\le} b_2 + \eta = \text{OPT} + \zeta,$$

where (b) is obtained by following Algorithm 4, i.e., rent at shop 1 with $r_1 = 1$, and (c) follows that $\zeta \ge x - y_1$, i.e., $x \le \zeta + y_1 \le \zeta + b_2$. Therefore, we have $CR_3 < 1 + \frac{\zeta}{\text{OPT}}$.

Finally, if $x \ge \left\lceil \frac{(m-2z+1)b_2}{\lambda} \right\rceil$, we have OPT $= b_2$, and

$$\begin{aligned}
\text{ALG} &= \left\lceil \frac{(m-2z+1)b_2}{\lambda} \right\rceil - 1 + b_1 \\
&\le \frac{(m-2z+1)b_2}{\lambda} + b_1 \\
&\overset{(d)}{\le} b_1 + \frac{m-2z+1}{m-2z+1-\lambda}\zeta \overset{(e)}{\le} b_1 + \frac{1}{1-\lambda}\zeta,
\end{aligned}$$

where (d) follows $\zeta \ge x - y_1 > \frac{(m-2z+1)b_2}{\lambda} - b_2$, i.e., $b_2 \le \frac{\zeta}{(m-2z+1)/\lambda-1}$, and (e) follows $m - 2z + 1 \ge 1$. Thus $CR_4 < \frac{b_1}{b_2} + \frac{1}{1-\lambda}\frac{\zeta}{\text{OPT}}$.

Combining $CR_1, CR_2, CR_3$ and $CR_4$, we have the first bound.

Now we prove the second bound. According to Algorithm 4, the skier rents the skis at shop 2 until day $\left\lceil \frac{\lambda b_2}{2z-m+1} \right\rceil - 1$ and then buys on day $\left\lceil \frac{\lambda b_2}{2z-m+1} \right\rceil$ at shop 2, when the predictions satisfy $z \ge m/2$. The corresponding cost is ALG $= r_2\left(\left\lceil \frac{\lambda b_2}{2z-m+1} \right\rceil - 1\right) + b_2$ when $x \ge \left\lceil \frac{\lambda b_2}{2z-m+1} \right\rceil$. It is easy to see that the worst competitive ratio is obtained when $x = \left\lceil \frac{\lambda b_2}{2z-m+1} \right\rceil$, for which we have OPT $= \left\lceil \frac{\lambda b_2}{2z-m+1} \right\rceil$. Therefore, we have

$$\begin{aligned}
\text{ALG} &= r_2\left(\left\lceil \frac{\lambda b_2}{2z-m+1} \right\rceil - 1\right) + b_2 \\
&\le \left(\frac{\lambda r_2}{2z-m+1} + 1\right) b_2 \\
&\le \left(\frac{\lambda r_2}{2z-m+1} + 1\right) \cdot \frac{2z-m+1}{\lambda} \cdot \left\lceil \frac{\lambda b_2}{2z-m+1} \right\rceil \\
&= \left(r_2 + \frac{2z-m+1}{\lambda}\right)\text{OPT} \le \left(r_2 + \frac{m+1}{\lambda}\right)\text{OPT},
\end{aligned}$$

where the last inequality follows $2z - m \le m$.

Similarly, the skier rents the skis at shop 1 until day $\left\lceil \frac{(m-2z+1)b_2}{\lambda} \right\rceil - 1$ and then buys on day $\left\lceil \frac{(m-2z+1)b_2}{\lambda} \right\rceil$ at shop 1, when $z < m/2$. The worst competitive ratio is obtained when $x = \left\lceil \frac{(m-2z+1)b_2}{\lambda} \right\rceil$ for which we have OPT $= b_2$, and

$$\begin{aligned}
\text{ALG} &= \left\lceil \frac{(m-2z+1)b_2}{\lambda} \right\rceil - 1 + b_1 \\
&\le \frac{(m-2z+1)b_2}{\lambda} + b_1
\end{aligned}$$

$$= \left(\frac{b_1}{b_2} + \frac{m - 2z + 1}{\lambda}\right) \text{OPT} \leq \left(\frac{b_1}{b_2} + \frac{m + 1}{\lambda}\right) \text{OPT},$$

where the last inequality holds since $m - 2z \leq m$.

## 7 Proof of Theorem 4

Here we consider four different cases.

*(1):* $z \geq m/2$ and $x \geq k$. It is clear that OPT $= \min\{b_2, x\}$. According to Algorithm 5, the skier should rent at shop 2 until day $j - 1$ and buy on day $j$. This happens with probability $p_i$, for $i = 1, \cdots, k$, and incurs a the cost is $(b_2 + (i - 1)r_2)$. We have

$$\mathbb{E}[\text{ALG}] = \sum_{i=1}^{k} (b_2 + (i - 1)r_2)p_i = \frac{r_2 k}{1 - (1 - \frac{r_2}{b_2})^k} \leq \frac{r_2 k/b_2}{1 - e^{-r_2 k/b_2}}b_2$$

$$\overset{(a)}{\leq} \frac{r_2 \frac{\lambda}{2z-m+1}}{1 - e^{-r_2 \frac{\lambda}{2z-m+1}}}b_2 \leq \frac{r_2 \lambda}{1 - e^{-r_2 \frac{\lambda}{m+1}}}b_2 \leq \frac{r_2 \lambda}{1 - e^{-r_2 \frac{\lambda}{m+1}}}(\text{OPT} + \zeta).$$

where (a) follows that $k \leq \frac{\lambda b_2}{2z-m+1}$, i.e., $k/b_2 \leq \lambda/(2z - m + 1)$ and $\frac{x}{1-e^{-x}}$ increases in $x \geq 0$.

*(2):* $y \geq m/2$ and $x \leq k$. We have OPT $= x$. If the skier buys the skis on day $i \leq x$, then it incurs a cost $(b_2 + (i - 1)r_2)$, otherwise, the cost is $xr_2$. Therefore, we obtain the robustness through the following

$$\mathbb{E}[\text{ALG}] = \sum_{i=1}^{x} (b_2 + (i - 1)r_2)p_i + \sum_{i=x+1}^{k} xr_2 p_i = \frac{r_2 x}{1 - (1 - \frac{r_2}{b_2})^k}$$

$$\leq \frac{r_2}{1 - e^{-r_2 k/b_2}}\text{OPT} \overset{(b)}{\leq} \frac{b_1}{b_2} \cdot \frac{r_2}{1 - e^{-r_2(\frac{\lambda}{m+1} - \frac{1}{b_2})}}\text{OPT}.$$

where (b) holds true since $k = \left\lfloor \frac{\lambda b_2}{2z-m+1} \right\rfloor \geq \frac{\lambda b_2}{2z-m+1} - 1$, i.e., $k/b_2 \geq \lambda/(m + 1) - 1/b_2$, and $b_1/b_2 > 1$. To get the consistency, we can rewrite the above inequality

$$\mathbb{E}[\text{ALG}] \leq \frac{r_2 \cdot k/b_2}{1 - e^{-r_2 k/b_2}}\text{OPT} + \frac{r_2 \cdot \zeta/b_2}{1 - e^{-r_2 k/b_2}}k \leq \frac{r_2 \lambda}{1 - e^{-r_2 \frac{\lambda}{m+1}}}(\text{OPT} + \zeta).$$

*(3):* $z < m/2$ and $x < l$. OPT $= \min\{b_2, x\}$. Similar to *Case 2*, we have

$$\mathbb{E}[\text{ALG}] = \sum_{i=1}^{x} (b_1 + (i - 1) \cdot 1)p_i + \sum_{i=x+1}^{l} x \cdot 1 \cdot p_i = \frac{x}{1 - (1 - 1/b_1)^l} \leq \frac{x}{1 - e^{-l/b_1}}$$

$$\leq \frac{x}{1 - e^{-(m-2z+1)/\lambda}} \leq \frac{r_2 \lambda}{1 - e^{-r_2 \lambda}}(\text{OPT} + \zeta) \leq \frac{r_2 \lambda}{1 - e^{-r_2 \lambda/(m+1)}}(\text{OPT} + \zeta).$$

*(4):* $z < m/2$ and $x \geq l$. OPT $= b_2$. Similar to *Case 1*, we have the robustness as

$$\mathbb{E}[\text{ALG}] = \sum_{i=1}^{l} (b_1 + (i - 1) \cdot 1)p_i = \frac{l}{1 - (1 - 1/b_1)^l} \leq \frac{l}{1 - e^{-l/b_1}} = \frac{\lceil \frac{z-2m+1}{\lambda}b_1 \rceil}{1 - e^{-l/b_1}}$$

$$\overset{(i)}{\leq} \frac{\frac{z-2m+1}{\lambda}b_1 + 1}{1 - e^{-(z-2m+1)/\lambda}} \leq \frac{\frac{b_1}{b_2}\frac{m+1}{\lambda} + \frac{1}{b_2}}{1 - e^{-1/\lambda}}\text{OPT},$$

where (i) follows that $l = \lceil \frac{m-2z+1}{\lambda} \rceil \geq \frac{m-2z+1}{\lambda}$, i.e., $\frac{l}{b_1} \geq \frac{z-2m+1}{\lambda}$. Again, we rewrite the above inequality to get the consistency

$$\mathbb{E}[\text{ALG}] \leq \frac{l}{1 - e^{-l/b_1}} \leq \frac{r_2 \lambda}{1 - e^{-r_2 \lambda/(m+1)}}(\text{OPT} + \zeta).$$

# 8 Additional Experimental Results: Prediction from a Single ML Algorithm

We provide additional experimental results.

**Unbiased prediction errors.** We characterize the impact of $\Gamma$ with the two possible values $\Gamma = 3b_1$ and $\Gamma = b_1$. The corresponding results are presented in Figures 1 and 2 in the main paper for Algorithm 2 and Algorithm 3, respectively. Here we present the third option with $\Gamma = 0.8b_1$, as shown in Figure 1. We have the similar observations where small $\lambda$ shows better performance with low $\sigma$, while less trust should be put on the prediction when $\sigma$ is large.

Figure 1: Impact of unbiased prediction errors with $\Gamma = 0.8b_1$ under *(Left):* Algorithm 2; *(Right):* Algorithm 3.

**Biased prediction errors.** We consider the impact of biases on prediction errors.

We consider three possible values of $10, 20, 50$ for $\delta$. The performance of Algorithm 2, and Algorithm 3 with $\Gamma = 3b_1$ and $\lambda = 0.5$ are shown in Figure 5 in the main paper. Here, we also consider other values of $\lambda = 0.25, 0.75, 1$, and $\Gamma = b_1$. The corresponding results for Algorithm 2 and Algorithm 3 are shown in Figures 2, 3, 4 and 5. Similar observations can be drawn as given in the main paper: a smaller bias benefits the CR when the variance is small; when the variance is large, the impact of bias is significantly reduced.

Figure 2: Impact of biased errors on Algorithm 2 with $\Gamma = 3b_1$. *(Left):* $\lambda = 0.25$; *(Middle):* $\lambda = 0.75$; *(Right):* $\lambda = 1$.

Figure 3: Impact of biased errors on Algorithm 2 with $\Gamma = b_1$. (a) $\lambda = 0.25$; (b) $\lambda = 0.5$; (c) $\lambda = 0.75$; (d) $\lambda = 1$.

Figure 4: Impact of biased errors on Algorithm 3 with $\Gamma = 3b_1$. *(Left):* $\lambda = 0.25$; *(Middle):* $\lambda = 0.75$; *(Right):* $\lambda = 1$.

Figure 5: Impact of biased errors on Algorithm 3 with $\Gamma = b_1$. (a) $\lambda = 0.25$; (b) $\lambda = 0.5$; (c) $\lambda = 0.75$; (d) $\lambda = 1$.

# 9 Additional Experimental Results: Prediction from Multiple ML Algorithms

Similarly, we present additional experimental results for MSSR with multiple ML Algorithms. We vary the number of ML predictions from 1 to 8, and set the associated predictions to $x + \epsilon$, where $\epsilon$ is drawn from a normal distribution with mean $\delta$ and standard variation $\sigma$, and $\Gamma = b_1$. We investigate the impacts of $m$, $\lambda$ and $\delta$ on the performance.

**The impact of the number of predictions** $m$. We fix $\lambda$ and investigate the impact of the number of predictions $m$ on the performance. The results with $\lambda = 0.5$ is presented in Figure 7 (a) in the main paper. Here we provide results for $\lambda = 0.25, 0.75, 1$, as shown in Figure 6. We have the same conclusion: For unbiased prediction errors and fixed $\lambda$, if the prediction is accurate (small $\sigma$), increasing $m$ improves the competitive ratio, however, more predictions hurt the competitive ratio when prediction error is large.

Figure 6: CR of Algorithm 4 under unbiased errors with *(Left):* $\lambda = 0.25$; *(Middle):* $\lambda = 0.75$; *(Right):* $\lambda = 1$.

**The impact of the hyperparameter** $\lambda$. We fix $m$ and investigate the impact of the hyperparameter $\lambda$. on the performance. The results with $m = 5$ is presented in Figure 7 (b) in the main paper. Here we provide results for $m = 3, 8$, as shown in Figure 7. We have the same conclusion: less trust achieves better competitive ratio when the prediction error is large.

**The impact of biased errors** $\delta$. We fix $m$ and $\lambda$ to investigate the impact of biased errors on the performance. The results for $m = 3$ and $m = 8$ with $\lambda = 0.25, 0.5, 0.75, 1$ are presented in Figures 8 and 9. Same conclusions are observed: For fixed $m$ and $\lambda$, a smaller bias benefits the competitive ratio when the variance is small, while a larger bias achieves a smaller competitive ratio when the variance is large.

Figure 7: CR of Algorithm 4 under unbiased errors with *(Left):* $m = 3$; *(Right):* $m = 8$

Figure 8: Impact of biased errors under Algorithm 4 with $m = 3$ and (a) $\lambda = 0.25$; (b) $\lambda = 0.5$; (c) $\lambda = 0.75$; (d) $\lambda = 1$.

Figure 9: Impact of biased errors under Algorithm 4 with $m = 8$ and (a) $\lambda = 0.25$; (b) $\lambda = 0.5$; (c) $\lambda = 0.75$; (d) $\lambda = 1$.

We also numerically evaluate the performance of the randomized algorithm (Algorithm 5) with predictions from multiple ML algorithms. As illustrated in Figure 10, with a fixed trust on the prediction (e.g., $\lambda = 0.5$), increasing the number of predictions $m$ can benefit the CR with small prediction errors (small $\sigma$). However, it is not always beneficial when the prediction is non-accurate (with large $\sigma$). It will be interesting but a daunting task to investigate the optimality in terms of $m$, $\lambda$ and $\sigma$ for the randomized algorithm. Similarly, we characterize the impact of the hyperparameter under a given number of predictions (e.g., $m = 5$) as shown in Figure 11. Again, we observe that more trust (small $\lambda$) will benefit the algorithm when the prediction is accurate, while less trust achieves better performance when the prediction error is large.

**Real-world dataset.** Finally, we evaluate the performance of Algorithm 4 using real-world data. We assume there are 3 predictions in total. These three predictions are drawn from 4 ML algorithms, one is from a prefect prediction, and the other three are predictions with errors as discussed in the main paper. We present here for completeness. First, we generate a similar distribution on the number of episodes watched by viewers for the season 11, and randomly draw the prediction $y$ from that distribution. We call this "Prediction 1". We then generate two other ("bad") predictions where "Prediction 2" follows that $y = 24 - x$, and "Prediction 3" satisfies $y = 1$ if $x \geq b_2$ and $y = 24$ otherwise.

We consider four cases: (i) *Case 1*: All three predictions are perfect; (ii) *Case 2*: two predictions are perfect with the third one from "Prediction 1"; (iii) *Case 3*: one perfect prediction, along with two bad predictions from "Prediction 1" and "Prediction 2"; and (iv) *Case 4*: three bad predictions

Figure 10: CR of Algorithm 5 for unbiased errors for $\lambda = 0.5$ with different $m$.

Figure 11: CR of Algorithm 5 for unbiased errors for $m = 5$ with different $\lambda$.

from "Prediction 1", "Prediction 2" and "Prediction 3". From Figure 12, we observe that (1) if prefect predictions are the majority, the performance will be significantly good; if not, increasing the number of good predictions will benefit the result. (2) when we put more trust on the predictions ($\lambda \to 0$), multiple bad predictions will do harm to the CR; when $\lambda \to 1$, the gap between multiple good and bad predictions will be narrowed, all four curves will show better or close performance than best deterministic algorithm without predictions.

Figure 12: CR of Algorithm 4 with 3 predictions using real-world dataset.