[Reviews · NeurIPS 2020]

Review 1

Summary and Contributions: The paper proposes an online algorithm that utilizes predictions made by a machine learning model for a generalization of the classical ski rental problem called multi-shop ski rental. In the multi-shop ski rental problem, there are ‘n’ ski shops each of which provides the option of renting skis for $r_i or buying skis for $b_i. The skier needs to decide which shop to choose and whether to rent or buy. Importantly, the skier must choose the shop upfront and then continue renting (or buying) from that same shop. Clearly, this generalizes the classical ski rental problem which is a special case with a single shop. In the learning augmented framework, the algorithm receives a prediction ‘y’ regarding the number of days the skier wishes to ski. The paper generalizes the results of [17] to the multi-shop setup and obtains consistent and robust online algorithms (deterministic and randomized). The paper also considers a setup with multiple predictions.

Strengths: The paper studies a natural generalization of the ski rental problem and generalizes the learning-augmented online algorithm for ski rental for the more general multi-shop ski rental problem. The paper is also well-written and includes significant experimental evaluation as well of the proposed algorithms.

Weaknesses: While it has natural applications, the multi-shop ski rental problem seems to be a minor generalization of the standard ski rental problem and essentially the same algorithms (with minor adaptations) yield optimal results. I find the paper lacking in technical novelty since the algorithms are minor variants of those for the standard ski rental problem. In the setup with multiple predictions, the error is defined to be the error of the \emph{worst} prediction unlike prior work [4] that obtains competitive ratios in terms of the error of the \emph{best} prediction (which can of course be much lower).

Correctness: Yes

Clarity: Yes

Relation to Prior Work: Yes

Reproducibility: Yes

Additional Feedback: Other comments: The MSSR problem mandates that the skier first chooses a shop and then must rent / buy at that same shop for the entire instance. Without this restriction the problem boils down to the standard ski rental problem with rent cost r_i and buy cost b_n since the algorithm can always rent from shop 1 and buy from shop n. It will be good to emphasize the importance of this restriction more in the model. In Lemma 1, the best deterministic online algorithm can choose any of the n shops depending on the argmin of specified term. However, both the randomized and deterministic algorithms that utilize the predictions only buy from either the first shop or the last shop. Isn’t it possible to obtain better robustness / consistency tradeoff by considering these other shops as alternatives? Line 193 is incorrect. As currently described even with lambda = 1, Algorithm 2 does not equal BDOA and in particular does not completely ignore the predictions. It still looks at the prediction to decide which shop to buy from. This is related to the comment above - it would be nice if Algorithm 2 reduced to BDOA for lambda = 1. Also Fig 1 clearly shows that the competitive ratio of the algorithm with lambda =1 does depend on the error.


Review 2

Summary and Contributions: The authors propose online algorithms using ML predictions for multi-shop ski-rental problem which is a generalization of the classic ski-rental problem where the decision problem is restricted to buying or renting the ski equipment from a single store. The consider all the special cases where the predictions come from a single predictor as well as multiple ML models. They present empirical evaluation of their algorithms for different values of hyperparameters.

Strengths: The paper considers all the important cases of the problem and presents both deterministic and randomized algorithms for the problem. The theoretical analysis is complete and clear. The empirical evaluation is also complete.

Weaknesses: One of the drawbacks of the model is the assumption of the skier having to transact with one store after selecting a store. Even though the authors give an example of cloud service providers, I believe each shop offering multiple levels of service might also induce the customer to choose between each level on different days. It'll help to distinguish from such a setting. After reading the rebuttal, I think addressing the extensions MSSR-S/E might help in strengthening the contributions of the paper. The authors did not offer a single proof in the main paper and moved all technical analysis to the supplementary material. Presenting proof sketches will help the reading.

Correctness: Though I did not read all the proofs carefully, they look correct.

Clarity: The paper is a little dense in results and sparse on details in the technical analysis section.

Relation to Prior Work: The work clearly compares with the previous work.

Reproducibility: Yes

Additional Feedback:


Review 3

Summary and Contributions: This paper considers the multi shop ski rental problem in the online algorithms with advice model. This is a popular emerging model in the intersection of algorithms and machine learning. The idea is that an online algorithm is given access to a prediction. Then the algorithm can use the prediction to make decisions. The goal is to bound the competitive ratio in terms of the error of the prediction. Recently, there has been a paper in this model addressing the ski rental problem. Tis paper considers the multi-shop ski rental problem. The paper shoes consistency and robustness results for the model. These follow along the same lines as prior work. Overall, I find that the algorithm and analysis are applications of the ideas that are used in prior work. They were also fairly simple even in the first paper addressing this. For this reason, I find the results good to know, but not particularly exciting. The reason to consider accepting the paper is the contribution to the growing interest in this line of work. It does indeed contribute to this theory. For these reasons, I believe this is a borderline paper.

Strengths: contributions to theory of online algorithms with predictions. Extends prior work and makes it more robust

Weaknesses: Algorithmic and analysis contributions are extensions not particularly novel

Correctness: I believe the claims are correct

Clarity: It is reasonably well-written

Relation to Prior Work: the paper does a good job overviewing the area

Reproducibility: Yes

Additional Feedback:

[Author Response · NeurIPS 2020]

We thank all reviewers for their valuable comments and suggestions. We respond to each comment below.

**Contribution, novelty and technical strength.** This section mainly responds to common concerns raised by reviewers.
The MSSR model seems like a "minor" generalization of the standard ski rental; however, MSSR requires the skier to
make a *two-fold* decision, i.e., *when* to buy and *where* to buy. This allows more heterogeneity in skier's options and
makes MSSR a more challenging problem that requires new design of online algorithms to achieve desired performance,
especially with ML Advice. Finally, MSSR is a more general modeling framework for online algorithm design than the
standard ski rental problem. Many new applications (see the supplementary material) can be modeled with MSSR.

While we follow the recent direction of online algorithms design with ML advice (e.g., Lykouris et. al., ICML 2018,
Purohit et al. NeurIPS 2018, etc.), the MSSR requires us to design online algorithms by carefully crafting the specific
structure of the problem which are different from existing literature, with a substantially different set of proof techniques.
In particular, we would like to highlight that the probability distribution functions for the randomized algorithm is
carefully designed to achieve a solid performance guarantee in terms of robustness and consistency metrics. Though the
structure is used in previous work, we believe that our randomized algorithm analysis with new distributions provides
the robustness and consistency results in a more systematic manner. Further, at the beginning of Section 3.3, we
emphasize that a straightforward extension of the existing distribution used for standard ski-rental fails to guarantee
solid robustness and consistency. This observation further clarifies the significance of the theoretical contribution.

Finally, as for the significance of the proposed algorithms in practice, our experimental results in Figure 6 demonstrate
that the online algorithms with ML advice can effectively resolve major drawbacks of classic online algorithms. In
other words, although online algorithms with ML advice and certain hyperparameter may not outperform pure online
algorithms, it is always possible to find the right hyperparameter such that the performance of online algorithm with
ML advice is better. This further demonstrates the importance and novelty of our proposed online algorithms with ML
advice in the sense that the ML predictions needs to be incorporated in a judicious manner.

**Assumption on the model.** This section mainly responds to concerns raised by Reviewers #2 and #3. In this paper,
we consider the basic setting of MSSR where the skier must choose one shop immediately after she starts the skiing and
must rent or buy the skies in that particular shops since then. This basic setting occurs in many real-world applications,
e.g., cloud computing systems as explained in the footnote on page 3.

Beyond this basic setting, there are several extensions of MSSR that can be considered, e.g., (i) MSSR with *switching*
*cost* (MSSR-S), i.e., the skier is able to switch from one shop to another at some non-zero costs; and (ii) MSSR with
*entry fee* (MSSR-E), i.e., there is an entry fee for each shop and no switching is allowed. In MSSR and MSSR-E, the
skier needs to answer two questions *at the very beginning*: *where* to rent or buy, and *when* to buy the skis. In MSSR-S,
the skier is able to decide where to rent or buy the skis *at any time*. It can be argued that MSSR-S and MSSR-E can be
equivalently reduced to MSSR, e.g., switching happens only when buying (Ai et. al. 2014, see [1] in the paper). To that
end, our proposed online algorithms with ML advice can be extended to these models with some minor changes in the
constant terms. We will add discussions on this generalization in final version.

**Lemma 1 and Algorithm 2.** This section mainly responds to concerns raised by Reviewer #2. In this paper, we focus
on the basic setting of MSSR and design the corresponding learning-aided online algorithms with desired performance
guarantee. The key motivation is two-fold: (1) to keep the core competency of online algorithms, i.e., performance
guarantee against worst-case, which is characterized by *robustness*; and (2) to achieve a provably improved performance
if the accuracy of ML-predictor is satisfactory, which is characterized by *consistency*. The hyperparameter $\lambda \in (0, 1)$ is
a design parameter that determines the confidence level of the ML-predictors in the online algorithm. This provides a
more powerful approach compared to traditional competitive algorithms that either does not rely on prediction at all, or
*fully* rely on the predictions. With proper tuning of $\lambda$, one can achieve the "best of both worlds" paradigm with $\lambda \to 0$
represents full trust on ML and $\lambda \to 1$ indicates no trust at all.

Under this algorithmic framework, BDOA is the best deterministic online algorithm in the basic setting. Since our
algorithm is evaluated using two criteria, i.e., robustness and consistency, one interesting problem is to investigate the
optimality of an algorithm, which leads to the consideration of the Pareto optimality. In other words, if an algorithm A is
Pareto-optimal, then there is no other algorithm that can achieve a better consistency (resp., robustness) than A without
sacrificing the robustness (resp., consistency). We did not claim the Pareto optimality of our proposed online algorithms.
Investigating their Pareto optimality or designing new online Pareto optimal algorithms with better robustness and
consistency tradeoff is an interesting future direction. However, our proposed algorithms exhibit desired consistency
and robustness performance, and are evaluated via extensive numerical evaluations including real-world datasets. To
that end, Algorithm 2 does not fully equal to BDOA as $\lambda \to 1$ in our algorithm design. As $\lambda \to 1$, we reduce the impact
of ML predictions on the algorithm to the minimum. Algorithm 2 can be easily generated to BDOA at $\lambda = 1$ with a
minor change in the algorithm description. We apologize for this confusion and will make it clear in final version.

**Proof sketch.** We will add and clarify the proof sketch to highlight the technical contributions. Thanks Reviewer #3.

[Meta-Review · NeurIPS 2020]

This paper builds on prior work for the ski rental problem using predictions. There is interest in this model of online algorithm analysis in the NuerIPS community. However, the paper does not give a lot of new technical insights. I believe this is a borderline paper. It offers a marginal improvement on an area of interest.